# Materials Used for the Microencapsulation of Probiotic Bacteria in the Food Industry

**DOI:** 10.3390/molecules27103321

**Published:** 2022-05-21

**Authors:** Ewa Kowalska, Małgorzata Ziarno, Adam Ekielski, Tomasz Żelaziński

**Affiliations:** 1Department of Technology and Food Evaluation, Institute of Food Sciences, Warsaw University of Life Sciences, 159c Nowoursynowska St., 02-776 Warsaw, Poland; malgorzata_ziarno@sggw.edu.pl; 2Department of Production Engineering, Warsaw University of Life Sciences, 02-776 Warsaw, Poland; adam_ekielski@sggw.edu.pl (A.E.); tomasz_zelazinski@sggw.edu.pl (T.Ż.)

**Keywords:** microencapsulation, probiotics, viability

## Abstract

Probiotics and probiotic therapy have been rapidly developing in recent years due to an increasing number of people suffering from digestive system disorders and diseases related to intestinal dysbiosis. Owing to their activity in the intestines, including the production of short-chain fatty acids, probiotic strains of lactic acid bacteria can have a significant therapeutic effect. The activity of probiotic strains is likely reduced by their loss of viability during gastrointestinal transit. To overcome this drawback, researchers have proposed the process of microencapsulation, which increases the resistance of bacterial cells to external conditions. Various types of coatings have been used for microencapsulation, but the most popular ones are carbohydrate and protein microcapsules. Microencapsulating probiotics with vegetable proteins is an innovative approach that can increase the health value of the final product. This review describes the different types of envelope materials that have been used so far for encapsulating bacterial biomass and improving the survival of bacterial cells. The use of a microenvelope has initiated the controlled release of bacterial cells and an increase in their activity in the large intestine, which is the target site of probiotic strains.

## 1. Introduction

For many years, scientists have overlooked the subject of the human digestive tract, specifically the intestines and intestinal microbiota. Only the rising interest in probiotic therapy and the effect of individual probiotic strains on various diseases of the digestive system have prompted researchers to study the environmental conditions in the intestines. The intestines contain numerous microorganisms which range from bacteria to primitive archaea and play a specific role in the guts. Among the microorganisms in the intestines, the most abundant are the bacteria of the genera *Bacteroides*, *Prevotella*, and *Ruminococcus*. The available literature indicates that the human intestines are colonized by over 1000 bacterial species, which together may constitute around 1–3 kg of body weight. Some of these species have the ability to cause diseases or serious disorders of the digestive system, while the vast majority perform many important functions enabling proper digestion and absorption of nutrients. The main tasks of the intestinal microbiota are to (1) produce short-chain fatty acids, vitamins, and metabolites; (2) break down nutrients; and (3) protect the body against pathogens and neutralize potential mutagens. As the gut microbiota contribute to maintain intestinal homeostasis and the health of the host, methods aimed at modifying the composition and function of the gut microbiota have been gaining popularity. Of these, the most common is probiotic therapy, which targets a specific health condition of a patient by reducing or eliminating the symptoms associated with the disease [1].

Due to the growing trend of probiotic use on the global market, as well as scientific reports regarding the real effect of probiotic bacterial strains in many diseases and conditions, it is important to improve the process of bringing probiotics to their place of action. We know from the available literature that the effects of probiotics are much broader than indicated by scientific reports generated a few years ago. Probiotics are most often used in ailments such as abdominal pain, diarrhea, vomiting, indigestion, or irritable bowel syndrome, i.e., indisposition of the digestive system. However, not much is being said about the use of certain probiotic strains in diseases, such as type 2 diabetes, cardiovascular diseases, or Alzheimer’s disease, where they can have a large impact on reducing the related symptoms or even significantly improving the patient’s health.

Many factors contribute to the development of Alzheimer’s disease, such as lifestyle, diet, cognitive abilities, education, past diseases and infections, inflammation, and even the quality and duration of sleep [2]. Many scientists and doctors point to a bad diet as the main cause of Alzheimer’s disease. The Mediterranean diet is very helpful in preventing the development of Alzheimer’s symptoms, as it assumes a high consumption of vegetables, fruits, and fatty fish, which provides people with a large amount of antioxidants, vitamins, and omega-3 and omega-6 fatty acids. These are dietary components that quite strongly modify the composition and functionality of the intestinal microbiota, which, as research shows, also influences the development of Alzheimer’s disease. Studies conducted on patients with Alzheimer’s disease clearly show reduced microbial diversity in the large intestine, as well as a decrease in the number of *Firmicutes* and an increase in *Bacterioidetes* [3]. These results were confirmed by a study on Chinese people with mild symptoms of initial Alzheimer’s disease, who had a lower diversity of bacteria in their stools than in healthy people [4]. This shows the great potential of probiotics as a tool for modulating the intestinal microbiota, and even for the development of a new method of treating Alzheimer’s disease [5].

Very similar mechanisms of action for probiotics have been proven in the prevention of Parkinson’s disease, which is the second most common neurodegenerative disorder after Alzheimer’s disease. The key to reducing the risk of Parkinson’s disease is to ensure that the gut microbiome is sufficiently diverse. The supply of probiotics will tighten the intestinal barrier and thus prevent bacterial translocation and inflammation of the nervous system. Particularly good results have been obtained using bacteria of the genus *L. acidophilus* and *B. infantis* [6].

Autism spectrum disorders are neurodevelopmental disorders that cause cognitive impairment. Through continuous communication along the brain–gut axis, the intestinal microbiota transmits information that affects the regulation and functioning of the central nervous and neuroimmune systems. Disturbances in this path or other unfavorable changes may cause neurological disorders. Therefore, microbiological balance and diversity are extremely important, as they ensure that the functions of the intestinal microbiota are undisturbed [7]. A study was carried out in which patients with autism spectrum disorders took a probiotic mixture containing *Lactobacillus delbrueckii* subsp. *bulgaricus*, *L. acidophilus*, *B. breve*, *B. longum*, *B. infantis*, *L. paracasei*, *L. plantarum*, and *S. thermophilus* for 4 weeks. In a 12-year-old boy with severe disorders, in addition to noting an improvement in digestive system function, a reduction in autistic core symptoms and the Autism Diagnostic Observation Schedule (ADOS) score were also recorded [8].

Probiotics also play a very important role in minimizing the risk of civilization diseases, such as cancer, depression, diabetes, and cardiovascular diseases. In diseases such as cancer, we should be very careful in administering probiotics as they have an immunomodulatory effect. This may not always be desirable in the treatment of cancer as we do not know if it will lead to the proliferation of cancer cells. However, clinical trials have mainly shown a beneficial effect of inhibiting the spread of cancer cells and the formation of metastasis. The strain that showed the best results was *L. fermentum* RM28 [9]. Probiotics help to minimize the occurrence of civilization diseases by reducing the severity of inflammation. Colonization of the large intestine with a large number of beneficial probiotic bacteria leads to an increase in the production of butyrate, which, in addition to its anti-inflammatory effect, also seals the intestinal epithelium and prevents harmful metabolites from entering the blood [10].

Unfortunately, it is difficult to deliver the appropriate number of live cells of probiotic strains to the large intestine, which is the target site. When bacteria pass through the digestive tract, they are exposed to factors that can weaken their action, such as hydrochloric acid or digestive enzymes [11,12,13] (Figure 1). As a result, ensuring the optimal supply of probiotic strains (10^8^–10^9^ colony-forming units [cfu]), which is necessary to induce the desired therapeutic effect, is challenging [14]. Over the years, several techniques have been proposed for probiotic microencapsulation, which is the best way to protect live bacterial cells during their gastrointestinal transit [15]. Microencapsulation not only secures the material inside the shell, but also enables its controlled release at a specific site. This process has been used to protect vitamins (mainly ascorbic acid and B-group vitamins), oils rich in omega-3 and omega-6 acids, microelements, essential oils, and plant extracts [16]. The present review summarizes the research results that have been obtained so far related to materials used in the microencapsulation of probiotic bacterial strains.

## 2. Probiotics

The concept of probiotics has been known for several decades, and its definition has changed over time with the emergence of new scientific reports. According to the most recent definition, probiotics are “live microorganisms which, when administered in sufficient quantity, confer health benefits to the consumer” [17]. Probiotics are widely used in the pharmaceutical industry due to their ability to improve health conditions (including mental state) and secondary functions (including vitamin synthesis and a reduction in the levels of cholesterol and the risk of colorectal cancer). Research shows that strains of the genera *Lactobacillus*, *Bifidobacterium*, and *Saccharomyces* have therapeutic effects (Table 1). However, these microorganisms should not be considered when using probiotics as the only components of dietary supplements or functional food additives. Strains of the genera *Bacillus* and *Pediococcus* have been recognized as potential probiotics [18].

Yeasts such as *Saccharomyces cerevisiae* and *Saccharomyces boulardii* are already available in commercial probiotic preparations for people suffering from traveler’s diarrhea or viral infections of the digestive system. The literature also indicates the possibility of using *Akkermansia muciniphila* and *Faecalibacterium prausnitzii* bacteria as probiotics [19]. Both of these species are found in the human intestine and are strict anaerobes, which makes them very sensitive to external factors (mainly oxygen) and difficult to enclose in a microcapsule. Therefore, appropriate materials and techniques are needed for the microencapsulation of probiotic bacteria that may be beneficial to health [20]. Unfortunately, many probiotic bacteria cannot pass through the initial section of the human digestive tract in a good condition. This is due to the specific structure of selected bacterial strains. Bacterial cells have effector molecules known as functional ligands, which are particularly adversely affected by the low pH in the stomach, as well as by the action of pepsin, bile salts, and proteases [21]. Therefore, in order to develop an effective probiotic preparation, it is critical to protect the bacterial cells during gastrointestinal transit using an appropriate material [22].

## 3. Materials Used for Encapsulating Probiotics

Many methods have been developed for microencapsulating probiotic bacteria in different materials. The most used materials are whey and soy proteins, alginates, pea proteins, acacia, pectins, chitosan, and carrageenans (Figure 2 and Figure 3, Table 2).

### 3.1. Whey Proteins

Whey proteins were the first to be used in the microencapsulation of probiotics. The natural environment for lactobacilli is fermented milk products, as well as products resulting from the fermentation process. Due to their chemical and physical properties, milk proteins, such as whey protein isolate obtained as a by-product of cheese production, are readily used in the production of probiotic microcapsules. Whey protein isolate is a rich source of functional proteins (90–96%), and its physicochemical properties allow for intermolecular cross-linking with other polymers [50]. Whey proteins are often added to carbohydrate biopolymers to increase the stability of microcapsules [51].

The preparation of whey proteins for the formation of microcapsules consists of dissolving the industrial whey protein isolate in deionized water and thoroughly mixing it with sodium hydroxide. The resulting solution is heated to 70 °C and then cooled to room temperature to prevent protein aggregation. Microcapsules based on whey proteins are mostly prepared using techniques that do not require the use of high temperatures capable of adversely affecting the structure of the final product [52]. A frequently used technique for the formation of microcapsules is extrusion, which uses an encapsulator and a calcium chloride solution to harden the outer shell [53].

Doherty et al. investigated the properties of whey-protein-based microcapsules containing *Lactobacillus rhamnosus*, i.e., one of the best known bacterial strains, as the core. The authors exposed the microcapsules to a low pH, which is comparable to that of the stomach (3.4, 2.4, 2.0), and obtained satisfactory results. They found that, compared to unprotected bacterial cells, the microcapsules survived adverse external conditions at an amount of 5.7 and 5.1 log cfu/mL. Whey-protein-based microcapsules showed good resistance to gastric acid, initiating the controlled release of live bacterial cells at the target site, i.e., the large intestine [22].

Microencapsulated probiotics are often added to food products that are more likely to be eaten by consumers than dietary supplements. These are usually fermented products such as kefir or dairy desserts (ice cream). Although these products serve as a good environment for probiotic bacteria, they may cause a reduction in their viability. Native bacterial cells of probiotic strains exhibit poor survival in products such as yogurt, but materials such as whey protein isolate microcapsules used for the microencapsulation of *Lactobacillus acidophilus* La-5 can help. It was found that microencapsulation increased the number of viable bacterial cells reaching the large intestine from 10^4^ to 10^6^ cfu/mL after consumption of yogurt stored for 10 weeks. Moreover, the addition of microencapsulated bacterial cells to yogurt did not cause a significant change in its pH or the initial bacterial microflora [35].

The addition of lignin material to protein capsules had a significant influence on the viability of *Lactobacillus reuteri* KUB-AC5 probiotic bacteria in the conditions of a simulated digestive system and during storage. Due to the interactions of lignin with whey proteins, the number of viable bacterial cells increased after refrigerated storage and the moisture content was reduced. The lowest and highest number of viable cells determined at the final stage of the experiment were 8.70 and 9.34 log cfu/mL, respectively. Lignin significantly improved the survival and structure of spray-dried bacterial cells, as well as the antioxidant properties of whey proteins (Figure 4) [45].

Lignin can be regarded as a promising material for agglomerates, allowing a stable habitat to be maintained for microorganisms, while ensuring their slow release into the environment [54].

### 3.2. Soybean Protein

Whey proteins offer many advantages in the microencapsulation of probiotics, but a significant disadvantage is that they cannot be consumed by certain people due to their allergenic properties. As a result, materials that do not cause adverse reactions when ingested are preferred for microencapsulation. These include soy protein isolates and other types of plant proteins, which have gained popularity in recent years. Soy protein isolates are widely available, have proven health and nutritional benefits, and are characterized by low immunogenicity and similarity to components of extracellular matrix tissue [55].

Most often, probiotic bacterial cells are microencapsulated in soy protein concentrate, which is produced from high-quality, clean, and husk-free soybeans. To extract the protein concentrate, the seeds are first processed into flour, from which the protein fraction can be easily obtained. The two main techniques applied with vegetable proteins for microencapsulation are spray-drying and extrusion. Due to the possibility of achieving better survival of probiotic bacteria, extrusion is increasingly used [56].

In some cases, a method is used for joining materials. This involves the use of calcium alginate and plant protein isolate (e.g., soybean), which are popular worldwide. The effectiveness of such a method was confirmed in a study by Hadzieva et al., in which the viability of *Lacticaseibacillus casei* 01 cells was evaluated after microencapsulation with a solution comprising calcium alginate and soy protein isolate using the spray-drying process [33].

In the cited study, the viability of bacterial cells after the encapsulation process was determined at 8.86–11.77 log cfu/g. Furthermore, with increasing alginate concentration, the viability of bacterial cells increased, and thus so did the effectiveness of microencapsulation. On the other hand, when the concentration of soy protein isolate was increased, the viability of bacterial cells decreased. However, when exposed to simulated gastric and intestinal juices, the number of viable bacterial cells did not drop below 10^7^ cfu/g. Microcapsules containing soy protein isolate also exhibited increased residence time in the small intestine and higher probability of bacterial cell release at the target site [33].

Interestingly, ready-made protein isolates are not preferred by all researchers for their experiments. In some studies, soy flour is subjected to extraction for obtaining soy proteins, which requires expertise and the use of reagents such as hexane or a carbonate buffer. In the study by Gonzalez-Ferrero et al., probiotic strains *Lactiplantibacillus plantarum* CECT 220 and *L. casei* CECT 475 were encapsulated in soy protein coacervates using a spray-drying process with calcium salts. The viability of bacterial cells was determined at 9.5–10.7 log cfu/mL before the microencapsulation process, while it was 9.6–10.2 log cfu/mL after microencapsulation. These results are in line with those of the above-cited study and confirm that soy proteins are a suitable material for the microencapsulation of probiotics [32].

The use of soy proteins for the microencapsulation of active ingredients, such as probiotics, has gained increasing interest. Compared to animal proteins, these vegetable proteins exhibit better gel-forming and emulsifying activity, better thermal stability, and resistance to mechanical stress [57].

### 3.3. Alginates

Alginates are widely used for microencapsulating probiotic bacterial strains. They are natural polysaccharides containing mannuronic and guluronic acid residues. Due to the sequential arrangement of these acid units, alginates in aqueous solutions have negative charges arranged along the skeleton, which allows them to form complexes with positively charged gelatin polymers. In this form, alginates are extremely stable at low pH; hence, they can effectively protect the active substances constituting the core of the capsule, while swelling under alkaline conditions to release probiotics and drugs in the intestinal lumen [58].

Microcapsule preparation involves several steps that will lead to the formation of a shell capable of protecting the core. If alginate is the material used for encapsulation, the first stage involves the formation of the first layer of the shell, mostly gelatin, which is only then cross-linked with alginate. The shell is created by thoroughly mixing all the materials that make up the microcapsule. To achieve this, gelatin is dissolved in NaCl solution at 50 °C. Subsequently, the cell suspension is added to aqueous gelatin and emulsified with vegetable oil for 30 min. In the next step, the resulting gelatin microspheres are coated with alginate. Then, external cross-linking is carried out using calcium ions. In short, the whole process involves thorough mixing of the microspheres with the alginate solution, filtration, and resuspension in oil and calcium ions to cross-link the alginate layer [59].

Mixing of microspheres with the alginate solution and calcium ions brings good results. To improve the microencapsulation process, some researchers have used encapsulators, which enable the production of microcapsules through injection pumps and special nozzles. The preceding stage, as described above, involves the thorough mixing of the cell suspension with a sodium alginate solution of appropriate viscosity, which allows the materials to combine [60,61,62].

Alginate microcapsules help to significantly improve the survival of Bifidobacterium bacteria. Research shows that alginate microspheres cross-linked with calcium ions showed better protection of bacterial cells during gastrointestinal transit. The number of viable nonencapsulated bacterial cells decreased by 3.45 log cfu/mL, while the number of viable cells encapsulated with alginate cross-linked with calcium ions decreased only by 1.75 log cfu/mL. When alginate microcapsules were exposed to simulated digestive juices, the number of viable bacterial cells was significantly higher, at a range of 7.35–7.57 log cfu/mL (Figure 5) [28].

As alginates are biocompatible with other polymers, they are often combined with other materials to obtain the most stable structures. The most used combinations are [27]:alginate and chitosan;alginate and calcium carbonate;alginate and gelatin;alginate and whey protein;alginate and oligosaccharides.

Some unconventional combinations, such as alginate and sea buckthorn, have also been used for microencapsulation. This combination has been used as a representative of functional foods and nutraceuticals. Adding these to structures, such as microcapsules, can significantly improve the health value of the final product. Sea buckthorn extracts are rich in unsaturated fatty acids and carotenoids, especially beta carotene, which serves as a food additive. Microcapsules based on alginate and sea buckthorn extract can resist temperatures up to 50 °C and maintain a survival rate of 6 log cfu/mL. Moreover, it was found that the survival of *L. casei* bacterial cells encapsulated using materials with sea buckthorn extract was 15% higher under simulated stomach conditions than when encapsulated using alginate alone [27].

### 3.4. Pea Protein

Legume proteins, including pea proteins, have also been used as an economic option for microencapsulation to improve the viability of bacterial cells. Some advantages of pea proteins include good water solubility, foamability, and high temperature stability. Pea protein isolates consist mainly of albumin and globulins; of these, the latter is more desirable because albumin contains many enzyme inhibitors and lectins, which may adversely affect the quality of end products. Another important advantage of pea proteins is their low cost. The cost of using these proteins in microcapsules is half that of using milk proteins and 25% less than that of using soy proteins. This is an advantage for scientists planning to conduct research with these materials, as it is often the cost that limits or prevents experimentation. Moreover, pea proteins have shown promising results in probiotic microencapsulation, both when used alone and in combination with polysaccharides [63].

Klemmer et al. [31] used pea protein isolate in combination with alginate for the microencapsulation of bifidobacteria. Initially, the authors tested the possibility of encapsulating bacterial cells with these materials separately. They observed that alginate could not provide sufficient protection to the cells when exposed to simulated gastric juice, while pea proteins alone were insufficient to maintain structural integrity during microencapsulation. On the other hand, the combination of these two materials helped to maintain the desired viability of bacterial cells during a 2 h incubation in simulated gastric and intestinal juice (Figure 6). Microcapsules of *Bifidobacterium* adolescentis, made using alginate and pea proteins in combination, were capable of gradually releasing bacterial cells over time [40].

Due to the effective encapsulation of probiotic bacteria, pea proteins have also been used for encapsulating probiotic yeast *S. cerevisiae* var. *boulardii*. Yeasts exhibit strong probiotic properties, and are often taken during “stomach flu” and travel to avoid diarrhea resulting from contact with foreign bacterial microflora. However, they are unstable during food processing and gastrointestinal transit. The microencapsulation of probiotic yeast with materials such as Arabic gum or pea protein isolate was shown to be effective and improved the survival of yeast cells when exposed to simulated gastrointestinal tract conditions. The obtained results confirmed the possibility of combining Arabic gum with pea protein isolate to increase the efficiency of encapsulation [64].

### 3.5. Arabic Gum

Arabic gum is a naturally occurring plant gum, which is derived from the trunk and branches of *Acacia Senegal* and other trees of the genus *Acacia* growing in Africa. It is used to encapsulate various active substances and probiotics due to its good water solubility, emulsifying properties, and low viscosity. For microencapsulation, Arabic gum does not require special preparation; it is obtained industrially and used directly in experiments. However, to optimize the microencapsulation of probiotic bacteria, it must be stabilized by materials such as gelatin, creating a sufficiently hard coating [65].

Optimum microencapsulation can be achieved with Arabic gum by applying techniques which do not adversely affect the shell material and coacervation, such as spray-drying [66].

Fazilah et al. reported the efficient encapsulation of *Lactococcus lactis* Gh1 using the spray-drying process and Arabic gum as a shell component, which helped to obtain powders with 10^9^ cfu/mL viable cells. The 2 h incubation of microencapsulated bacterial cells did not cause a significant decrease in the number of live cells, which was still high at 1.11 × 10^6^ cfu/mL [37].

The above results were confirmed in a study using probiotic cells of *L. plantarum*, in which gelatin and Arabic gum were used as materials for encapsulation. The viability of bacterial cells’ postencapsulation was satisfactory, at a level of 8.6 log cfu/mL, while the efficiency of the microencapsulation process was 97.78%. When the encapsulated and free cells were exposed to simulated digestive juices, 80% of the encapsulated cells survived while only 25% of nonencapsulated ones were viable [38].

### 3.6. Pectins

Pectins are nontoxic materials which can form gel structures when combined with divalent metal ions, such as Ca^2+^. The use of highly methoxylated pectins can result in the efficient encapsulation due to their high gelation strength [67,68,69]. Pectin-based microcapsules are most often prepared using the emulsion method. Briefly, the pectin solution is mixed with the cell suspension, and, then, the whole mixture is added to the vegetable oil and calcium chloride using a dropper. The interaction of pectins with calcium chloride results in hardening of the capsule wall [70].

Microcapsules made using pectin along with inulin obtained from Jerusalem artichoke encapsulated bacterial cells at a level of 96%. Furthermore, the viability of bacterial cells after 42 days of storage was satisfactory, amounting to seven logarithmic degrees [29].

Pectins have also been proven to be successful in combination with alginate. The encapsulation of *L. acidophilus* La-5 and *Bifidobacterium animalis* subsp. *lactis* BB-12 cells with these two materials resulted in much higher survival compared to the viability of nonencapsulated cells. Storage for 30 days caused a decrease in the number of viable bacterial cells, but the number did not drop below 10^6^ cfu/mL, which confirms the efficiency of encapsulation. The overall encapsulation efficiency was 92% [30].

### 3.7. Chitosan

Chitosan, a natural polysaccharide formed as a result of chitin deacetylation, has great potential for microencapsulating bacterial cells. An encapsulation system based on a mixture of chitosan and xanthan shows specific delivery and the controlled release of the microcapsule core at the target site. Due to ionic interactions, chitosan and xanthan form structures that are resistant to digestive enzymes and tolerate low pH [71]. Both xanthan and chitosan do not require special preparation steps prior to their use in microencapsulation. They are available industrially in a form that encourages their immediate use in research. The pH of these materials can be adjusted by dissolving them in appropriately concentrated HCl and adding deionized water. Chitosan is used in the final stage of microcapsule production. Its role is to harden the shell around the cell suspension, constituting the core of the capsule [43].

Chitosan and xanthan are often combined with extrusion, as its two-stage procedure can obtain the desired effects. The process is carried out using a syringe with a suitable cannula through which the xanthan mixture with the cell suspension is pressed into the chitosan hardening solution [72].

Compared to the previously described materials, the encapsulation efficiency achieved using microcapsules based on chitosan and xanthan was estimated at 86%. However, the survival of encapsulated bacterial cells added to yogurt and stored for 21 days under refrigerated conditions was found to be promising. Moreover, the addition of these materials improved the acid profile of yogurt [43]. Chitosan is widely used as an additive to alginate microcapsules, on which it is coated in order to strengthen the structure of the alginate gel. Microcapsules without chitosan coating have been found to be overexposed to gastric acid, which penetrates the surface holes of the alginate structure and reaches the core, reducing the viability of the bacterial cells [73].

Chitosan, in combination with calcium alginate, starch, or inulin, is an appropriate material for increasing the viability of probiotic bacterial strains. Chitosan coating was observed to protect cells when they were incubated in simulated gastrointestinal juices. Moreover, the diameter of the microcapsules increased significantly after the addition of the chitosan coating [44]. Although chitosan cannot be used alone for the microencapsulation of bacterial cells, it is a very good additive that can increase the survival of bacteria in the final product.

### 3.8. Carrageenans

Carrageenans are polysaccharides obtained from microalgae and red seaweed, and are commonly used as raw materials in the production of jellies and gels. They occur in three forms—kappa, lambda, and iota, with the former being the most popular. Like chitosan, carrageenans cannot be used individually in microcapsules. They are usually added as a component of the structure intended for encapsulation and most often combined with sodium alginate [74]. For the preparation of carrageenan-based microcapsules, a hardener solution, mainly calcium chloride, is often used. The techniques applied to produce carrageenan microenvelopes are extrusion (with the use of a syringe and cannula or a specialized extruder), emulsification, and internal gelling [75].

Afzaal et al. [41] investigated the encapsulation efficiency of carrageenan for bacterial cells and its influence on cell viability and stability in yogurt and simulated gastrointestinal conditions. On day 0, the number of probiotic cells encapsulated with sodium alginate–carrageenan mixture was determined at 9.89–9.91 log cfu/mL, but the number gradually decreased to 8.39–8.74 log cfu/mL. Uncoated cells added to yogurt showed very poor survival. Similarly, after incubation in simulated gastrointestinal juices, the survival of encapsulated probiotic bacterial cells was found to be higher than that of free cells [41].

As mentioned above, carrageenan capsules can effectively protect bacterial cells against harmful conditions, including those prevailing in the food products to which they are added. One such product is cheddar cheese, to which *Bifidobacterium bifidum* cells were added, and the encapsulated probiotic bacteria were found to show better survival compared to nonencapsulated probiotic cells. A log reduction of 2.60 cfu/g was observed in the viability of unencapsulated cells, while in the case of cells encapsulated with sodium alginate and kappa–carrageenan, the viability reduced by 1.03 and 1.48 log cfu/mL, respectively. Encapsulation with the abovementioned materials influenced the protection and stability of probiotic cells under unfavorable conditions [42].

### 3.9. Ethylcellulose

Ethylcellulose is a linear polysaccharide formed when the hydroxyl groups of cellulose are replaced with ethyl groups. Its unique properties allow for the effective microencapsulation of probiotic bacteria. The most important of these properties are:water insolubility;hydrophobicity;physiological indifference;lack of odor;lack of taste;low amount of calories;stability during storage.

Ethylcellulose is mainly used in commercial oral pharmaceuticals, in which it serves as a coating to control the release of active ingredients during gastrointestinal transit. Ethylcellulose shells have been used to encapsulate substances such as ketoprofen [76] and quercetin [77]. The technique that facilitates effective production of ethylcellulose-based microcapsules is electrospray, which requires the use of specialized pumps, needles, and a high-voltage generator.

Using the electrospray method, an attempt was made to encapsulate cells of probiotic bacteria *B. animalis* subsp. *lactis*. Moreno et al. used maltodextrin and glycerol to stabilize the core, which was the cell suspension. These materials in combination with ethylcellulose allowed the authors to obtain a large number of viable bacterial cells. After microencapsulation, the number of viable cells remained at 10^9^–10^11^ cfu/mL. The efficiency of microencapsulation with ethylcellulose was particularly evident when the number of viable bacterial cells after 4 weeks of storage at 30 °C was compared with that determined before storage. It was observed that cells coated with ethylcellulose remained viable at a level adequate to induce the desired therapeutic probiotic effect, whereas the viability of uncoated cells was reduced by 7.57 cfu/mL. Thus, the use of ethylcellulose can possibly increase the survival rate of probiotic bacteria [46]. So far, this is the only study to have confirmed the possibility of microencapsulating probiotics using ethylcellulose.

### 3.10. Starch

Starch is one of the largest natural polysaccharides that is easily modifiable and is therefore widely used in medicine and tissue engineering. Since its processing requires complex techniques, there is a need to develop more resistant derivatives for producing coatings to achieve a controlled release of active substances in the body [78].

Puttarat et al. microencapsulated *L. reuteri* TF-7 with a mixture of whey protein isolate and nanocrystalline starch using the spray-drying technique, and observed increased stability and survival of bacterial cells under various unfavorable conditions. The production of spherical microcapsules allowed the authors to obtain a larger number of viable bacterial cells after exposure to pH, heat, and gastrointestinal environment. Moreover, the probiotic cells retained their biological activity, and thus induced the desired therapeutic effect in patients who consumed the probiotic preparation [79].

Lancuški et al. presented an innovative approach for the microencapsulation of probiotics. They used starch formate and glycerol to suspend the cells and applied the electrospinning technique for encapsulation, which requires the preparation of special starch formate films or coatings. These coatings were performed in two stages: the solvent was evaporated in the first stage, and the coatings were baked in an oven at 105 °C in the next [47].

For the evaluation of bacterial cells, the authors used interesting methods based on calorimetry, thermomechanical analysis, and Fourier-transform infrared spectroscopy. It was observed that *Lacticaseibacillus paracasei* cells retained their stability and viability for a longer duration during refrigerated storage. This proved that starch fibers in combination with glycerol can serve as a material for effective microencapsulation of biotherapeutics [47].

### 3.11. Pullulan

Pullulan is an extracellular polysaccharide produced by *Aureobasidium pullulans*. Due to its safety and nontoxicity, this material has been used to produce coatings in the food and pharmaceutical industries. Pullulan is also tasteless, odorless, colorless, and has thermostable properties. Recently, it has been proposed that whey proteins can be combined with polysaccharides to produce edible films that can extend the shelf life of products and reduce moisture loss [80].

Two groups of researchers tested the interrelationship between pullulan and whey proteins in the context of microcapsule formation for probiotic lactic acid bacteria. They used these two materials and applied emulsification and cold-gelling techniques to obtain satisfactory results. It was noted that the number of encapsulated live cells of *L. acidophilus* NRRL-B 4495 strain decreased by only 1.64 log cfu/mL, while the number of nonencapsulated live cells decreased by less than 4 log cfu/mL [81]. When *L. rhamnosus* bacteria were encapsulated by combined emulsification and extrusion, the gels obtained from whey proteins and pullulan exhibited low solubility in simulated gastric juice, while the solubility was quite high in simulated intestinal juice. This enabled the controlled release of the probiotic during gastrointestinal transit [82]. Furthermore, it was shown that if materials such as whey proteins and pullulan are used separately, the cell suspension is much more effectively protected by protein shells than by polysaccharide shells. This effect is also influenced by the method used for encapsulation. For example, electrospraying, which involves the use of an apparatus equipped with a high-voltage power supply, showed better results in the case of protein shells [48].

### 3.12. Maltodextrins

Maltodextrins are easily digestible oligosaccharides used as food additives. They are made of D-glucose units and are formed from starch hydrolysis. Maltodextrins exhibit excellent solubility in water and low hygroscopicity. They are also nontoxic, odorless, and edible. All these features make them a perfect carrier for active substances and thus an ideal material for the microencapsulation of bacterial cells. Maltodextrins are produced from a variety of raw materials using different methods and processing conditions [83].

The material with which maltodextrins are often used in encapsulation is whey protein. In combination with maltodextrins, whey proteins form structures that effectively protect the core of the capsule and are susceptible to hardening in a calcium chloride solution [83]. For *L. acidophilus* La-5, a method of microencapsulation based on spray-drying with the use of soy extract and maltodextrin was proposed. Although the obtained results were not as promising as those obtained with the addition of whey proteins, the final encapsulation efficiency was estimated at 83% [84]. Libran et al. presented an innovative approach called electrospray coating atomization, which is used to increase the viability of freeze-dried bacterial cells. The authors coated bacterial lyophilisates with particles of food hydrocolloids using electrospraying. The process consisted of three stages and involved the application of a new layer of encapsulating materials each time. The best results were obtained with the use of whey protein concentrate and maltodextrin [49]. 

To sum up subject of microcapsules, below we show an examples of use of microencapsulated probiotic bacteria in food products (Table 3). 

## 4. Conclusions

In recent years, the food industry has been introducing a greater number of products containing probiotics. This is due to the growing awareness of consumers about healthy eating and the benefits of consuming probiotic bacteria. However, one of the main challenges faced by the food industry is ensuring that the products have an adequate number of viable bacterial cells on the shelves and that these cells are maintained during the storage period as they might have health effects on consumers. This problem is being addressed by researchers working on creating microcapsules that are biocompatible with bacterial cells. Materials and microencapsulation techniques should be appropriately chosen to ensure the best possible protection of bacteria without compromising the characteristics of the final product. It is difficult to adequately protect the core of the microcapsule (in this case, bacterial cells) from conditions prevailing in the digestive system, including digestive enzymes and pH changes. Previous studies on materials such as alginates and whey proteins have shown very promising results. However, methods using these materials are yet to be optimized. To adapt to the current trends in the food market, researchers have focused on developing microcapsules based on plant proteins that are less allergenic than milk proteins and are suitable for vegans and vegetarians. The efficiency of microencapsulating probiotics with plant matrices is comparable with that obtained using matrices of animal origin. Further research, along with modification of the methods developed so far and their optimization, is necessary to achieve the maximum viability of bacterial cells.

## Figures and Tables

**Figure 1 molecules-27-03321-f001:**
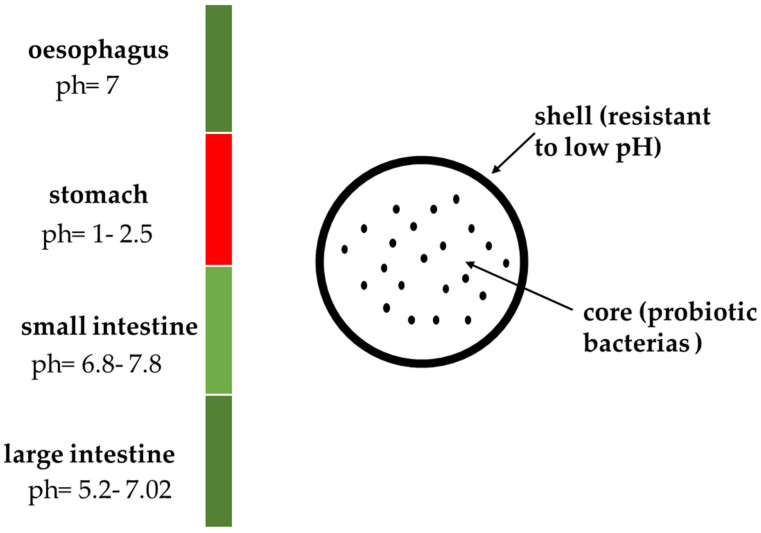
Graphic presentation of the conditions in the digestive system to which probiotic bacterias are exposed (own study).

**Figure 2 molecules-27-03321-f002:**
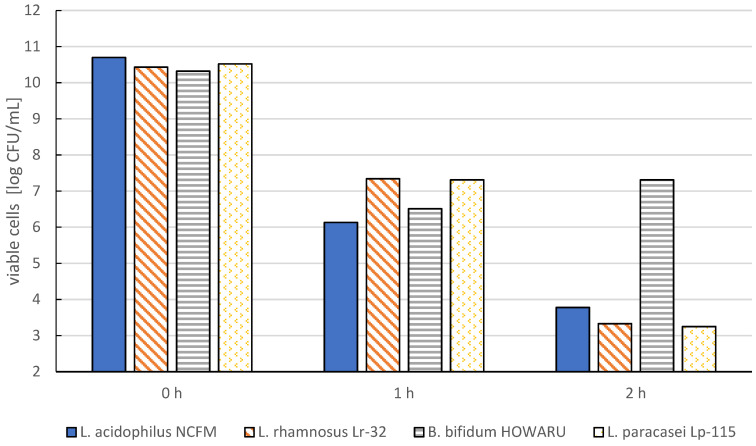
Impact of holding time in a low pH environment (pH value = 2) on the viability of the selected probiotic bacteria cells (based on Ding and Shah, 2009 [25]).

**Figure 3 molecules-27-03321-f003:**
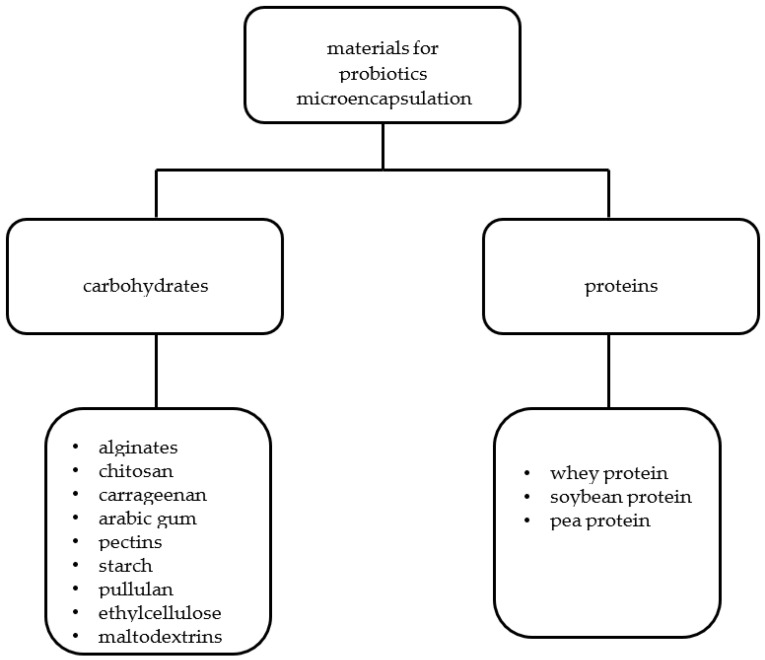
Division of materials used for the microencapsulation of probiotics (own study).

**Figure 4 molecules-27-03321-f004:**
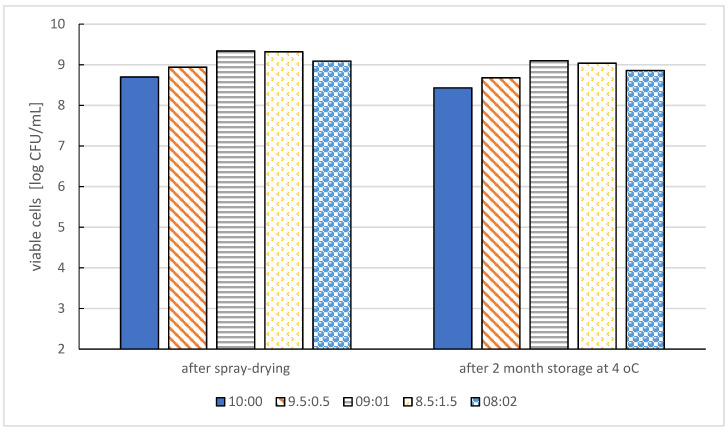
Viability of microencapsulated selected probiotic bacteria cells (L. reuteri KUB-AC5) by various whey protein isolate and lignin ratio [45].

**Figure 5 molecules-27-03321-f005:**
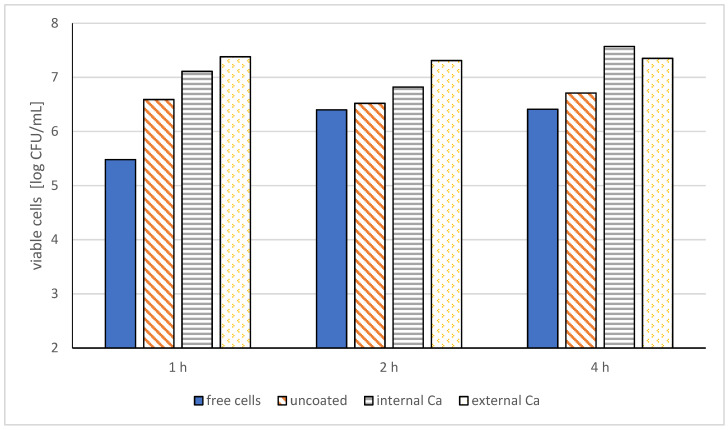
Viability of free and microencapsulated selected probiotic bacteria cells (*Bifidobacterium adolescentis* 15703T) by alginate-coated gelatin microspheres with internal and external Ca+ [28].

**Figure 6 molecules-27-03321-f006:**
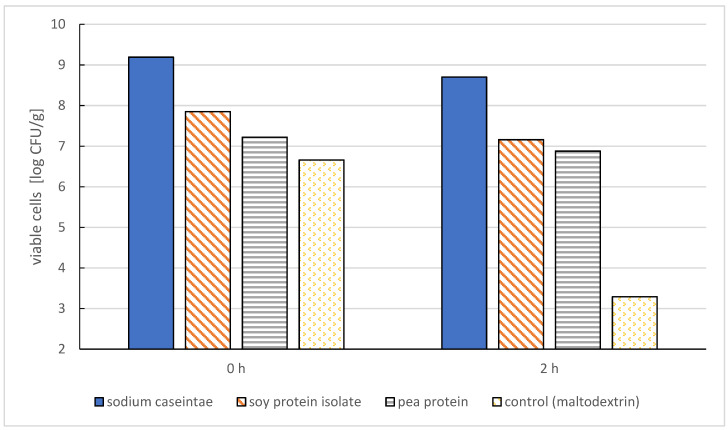
Viability of microencapsulated selected probiotic bacteria cells (Lactobacillus acidophilus La-5) after 2 h in simulated gastric juice [39].

**Table 1 molecules-27-03321-t001:** Examples of strains of the most commonly used bacteria and probiotic yeast [23,24].

Type	Probiotic Strain
*Bifidobacterium*	*Bifidobacterium animalis* ssp. *lactis* BB-12, *Bifidobacterium bifidum* Rosell- 71, *Bifidobacterium bifidum* W23, *Bifidobacterium infantis* DSM24737, *Bifidobacterium lactis* BB-12, *Bifidobacterium lactis* W52, *Bifidobacterium longum* W11, *Bifidobacterium adolescentis* NK 98
*Lactobacillus*	*Lactobacillus acidophilus* La-5, *Lactobacillus acidophilus* NCFM, *Lactobacillus acidophilus* LA 1, *Lactobacillus acidophilus* LA 14, *Lacticaseibacillus casei* Shirota, *Lacticaseibacillus casei* Rosell-215, *Lacticaseibacillus casei* CRL431, *Lactobacillus delbrueckii* ssp. *bulgaricus* Lb-87, *Lactobacillus helveticus* Rosell- 52, *Lactiplantibacillus plantarum* CECT7484, *Lactiplantibacillus plantarum* 299v, *Lacticaseibacillus rhamnosus* ATCC 53103, *Lacticaseibacillus rhamnosus* GR-1, *Lacticaseibacillus rhamnosus* GG, *Limosilactobacillus reuteri* MM53, *Limosilactobacillus fermentum* RC-14, *Lacticaseibacillus casei* 01, *Lactiplantibacillus plantarum* CECT 220, *Lacticaseibacillus casei* CECT 475
*Saccharomyces*	*Saccharomyces boulardii* CNCM I- 745, *Saccharomyces cerevisiae* var. *boulardii* DSM 27112
*Lactococcus*	*Lactococcus lactis* Gh1

**Table 2 molecules-27-03321-t002:** Materials used for the microencapsulation of probiotic bacteria.

Microcapsule Material	Bacterial Strain	Conclusions	References
alginate	*Lactobacillus acidophilus* CSCC 2400	There was an increase in the number of viable probiotic bacteria cells in alginate microcapsules	[26]
	*Lacticaseibacillus casei* ATCC 393	Microcapsules based on alginate and sea buckthorn extract are particularly resistant to high temperatures, up to 50 °C	[27]
	*Bifidobacterium adolescentis* 15703T	Compared to non-encapsulated bacterial cells, the number of cells encapsulated with alginate decreased by 1.75 log cfu/mL	[28]
pectins	*Lactobacillus rhamnosus* GG	The bacterial viability after the 42-day storage period was 7 logarithmic degrees	[29]
	*Lactobacillus acidophilus* La-5, *Bifidobacterium animalis* subsp. *lactis* BB-12	The bacterial viability after the 42-day storage period was 7 logarithmic degrees	[30]
soybean protein	*Lactobacillus bulgaricus* FTDC 1511	Soy microcapsules were characterized by a high protective effect	[31]
	*Lactoplantibacillus plantarum* CECT 220, *Lacticaseibacillus casei* CECT 475	The viability of bacterial cells before the microencapsulation process was between 9.5 and 10.7 log cfu/mL, after the microencapsulation process 9.6 and 10.2 log cfu/mL	[32]
	*Lacticaseibacillus casei* 01	The efficiency of encapsulating bacterial cells with soy proteins was confirmed	[33]
whey protein	*Bifidobacterium breve* R070, *Bifidobacterium longum* R023	Both strains after the microencapsulation process showed good tolerance to simulated gastrointestinal conditions	[34]
	*Lactobacillus acidophilus* La-5	The microencapsulation process increases the number of live bacterial cells reaching the large intestine from 10^4^ to 10^6^ cfu/mL	[35]
	*Lactobacillus rhamnosus* GG	The microcapsules survived exposure to adverse external conditions in the amounts of 5.7 and 5.1 log cfu/mL	[22]
prebiotics	*Lactobacillus acidophilus* ATCC 43121	Prebiotic microenvelopes did not induce a significant improvement in the survival of bacterial cells when exposed to low pH and high temperature	[36]
arabic gum	*Lactococcus lactis* Gh1	The use of Arabic gum as a shell component allowed authors to obtain powders with living cells at the level of 10^9^ cfu/mL	[37]
	*Lactiplantibacillus plantarum*	The viability of bacterial cells after encapsulation was 8.6 log cfu/mL, and the efficiency of the entire microencapsulation process was 97.78%	[38]
pea proteins	*Lactobacillus acidophilus* La-5	There was a decrease in the survival of bacterial cells by less than a logarithmic cycle, which confirmed the effectiveness of microencapsulation with pea proteins	[39]
	*Bifidobacterium adolescentis*	The desired viability of the bacterial cells was maintained during a 2 h incubation in simulated gastric and intestinal juice	[40]
carrageenans	*Lactobacillus acidophilus* ATTC-4356	Carrageenan microenvelopes did not significantly improve the survival of bacterial cells, despite 96% encapsulation efficiency	[41]
	*Bifidobacterium bifidum* ATTC-29521	The encapsulated probiotic bacteria showed a better survival compared to the non-encapsulated probiotic	[42]
chitosan	*Lactobacillus acidophilus*	Improving the acid profile of yogurt after adding microencapsulated bacterial cells	[43]
	*Lacticaseibacillus casei* ATCC 3939, *Bifidobacterium bifidum* ATCC 29521	Chitosan coating was an important factor in protecting cells during incubation in simulated gastrointestinal juices	[44]
lignin-whey protein	*Lactobacillus reuteri* KUB-AC5	The thermal stabilization and antioxidant properties of the proteins were enhanced by the lignin coating	[45]
cellulose	*Bifidobacterium animalis* subsp. *lactis (Bifido)* DSM 33443)	Cells coated with ethylcellulose remained viable at a certain level, allowing for a therapeutic probiotic effect; uncoated cells lost viability by 7.57 cfu/mL	[46]
starch	*Lactobacillus paracasei*	*Lacticaseibacillus paracasei* cells retained their stability and relatively long viability during refrigerated storage	[47]
pullulan	*Bifidobacterium animalis* subsp. *lactis* Bb12	Electrospray encapsulation significantly increased the viability of the bifidobacterial strain, especially at 20 °C	[48]
maltodextrins	*Bifidobacterium longum* subsp. *infantis* CECT 4552	Microencapsulation was most effective when maltodextrin was combined with whey proteins	[49]

**Table 3 molecules-27-03321-t003:** An example of the use of microencapsulated probiotic bacteria in food products.

Bacterial Strain	Material	Product	Reference
*Lacticaseibacillus paracasei* ssp. *paracasei* NFBC 338	milk powder	cheddar cheese	[85]
*Lacticaseibacillus paracasei* ssp. *paracasei* LBC-1e	sodium alginate	mozzarella cheese	[86]
*Bifidobacterium longum* B6	kappa- carrageenan	yoghurt	[87]
*Lactobacillus acidophilus* DD910 *Bifidobacterium lactis* DD920	alginate, resistant corn starch	yoghurt	[88]
*Lactiplantibacillus plantarum* TISTR 050	sodium alginate, soy protein isolate	pasteurized mango juice	[89]
*Bifidobacterium lactis* DSM 10140	gellan gum, xanthan gum	fermented African drink	[90]
*Lactobacillus acidophilus* La-5	pectins, calcium chloride, whey protein isolate	yoghurt	[91]

## Data Availability

Not applicable.

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
