# Peer review of "Materials Used for the Microencapsulation of Probiotic Bacteria in the Food Industry"

_molecules, 2022, doi:10.3390/molecules27103321_

Round 1

Reviewer 1 Report

This study described the various envelope materials that have been utilized before encapsulating bacterial biomass and improving bacterial cell viability. The usage of the microenvelope has allowed for the regulation of bacterial cell release and boosted their activity in the large intestine, which is the target site of probiotic strains. The manuscript is well structured and well discussed. However, some points should be checked and corrected before its acceptance in this journal. 

Therefore, according to my comments, I recommended the publication of the paper after major revision.

  • The study's background should be clearly stated. Describe the introduction and review of the work (Please add more information).
  • Please draw some good figure for easily understanding of the concept of review.
  • The MS English needs to be improved. The article's English must be carefully checked for grammatical errors.

Author Response

Thank you for your comments to improve the manuscript.

Reviewer 2 Report

The manuscript entitled “Materials used for microencapsulation of probiotic bacteria in the food industry.” aims to review the different types of envelope materials that have been used so far for encapsulating bacterial biomass and improving the survival of bacterial cells. This topic is highly important since there are meaningful indications that the human microbiome has an important role in many diseases, especially neurological. Regulation of human intestine flora may be the answer to many troubles, so to make the delivery systems for probiotics efficient is of uttermost significance. The manuscript is very well prepared. However, I have a few suggestions for improvement.

-The title should be without the period in the end.

-The authors could provide a short overview of the diseases and conditions currently considered to have the potential to be treated with probiotics, like Alzheimer’s disease, autism spectrum disorder, multiple sclerosis, Parkinson's disease, stroke, hypersensitivity reactions, autoimmunity, chronic inflammation, cancer etc.

-The references should be all formatted the same.

Some minor comments

Line 51: “therapeutic effects, is challenging”

Line 51: “have been proposed over the years for probiotic microencapsulation”

Line 64: “due to their ability to improve health conditions, including”

Line 74: “Both of these species are found”

Line 80: “It is due to the specific structure of selected bacterial strains.”

Line 132: “moisture content was reduced.”

Line 138: “habitat for microorganisms while ensuring their”

Line 154: “(e.g. soybean), which are popular worldwide”

Line 166: “intestine and a higher probability of bacterial”

Line 168: “soy flour is subjected to extraction to obtain soy”

Line 233: “used as an economical option”

Line 235: “and high-temperature stability. Pea”

Line 288: “Pectins are nontoxic materials that can form gel structures”

Line 294: “results in the hardening of the capsule wall”

Line 360: “the viability was reduced by”

Line 431: “This enabled the controlled release”

Line 441: “They are also non-toxic, odorless”

Line 459: “has been introducing more number of products”

Author Response

  • Author response to report 2:

Author’s notes

Thank you for your comments to improve the manuscript.

The title should be without the period in the end.

Thank you for your comment. We removed the period from the title.

The authors could provide a short overview of the diseases and conditions currently considered to have the potential to be treated with probiotics, like Alzheimer’s disease, autism spectrum disorder, multiple sclerosis, Parkinson's disease, stroke, hypersensitivity reactions, autoimmunity, chronic inflammation, cancer etc.

Thank you for your comment. We add a short overview of the diseases and conditions currently considered to have the potential to be treated with probiotics in the Introduction (lines 46-103).

The references should be all formatted the same.

Thank you for your comment. We improved and unified the citation style.

Line 51: “therapeutic effects, is challenging”

Line 51: “have been proposed over the years for probiotic microencapsulation”

Line 64: “due to their ability to improve health conditions, including”

Line 74: “Both of these species are found”

Line 80: “It is due to the specific structure of selected bacterial strains.”

Line 132: “moisture content was reduced.”

Line 138: “habitat for microorganisms while ensuring their”

Line 154: “(e.g. soybean), which are popular worldwide”

Line 166: “intestine and a higher probability of bacterial”

Line 168: “soy flour is subjected to extraction to obtain soy”

Line 233: “used as an economical option”

Line 235: “and high-temperature stability. Pea”

Line 288: “Pectins are nontoxic materials that can form gel structures”

Line 294: “results in the hardening of the capsule wall”

Line 360: “the viability was reduced by”

Line 431: “This enabled the controlled release”

Line 441: “They are also non-toxic, odorless”

Line 459: “has been introducing more number of products”

Thank you for your comments. Language errors have been corrected in linguistic proofreading proces.

Thank you for all your comments and suggestions. We hope, that the revised version of the manuscript is satisfactory.

Reviewer 3 Report

The review is well-conducted, despite only one of the authors has some papers on this matter (three old articles and not in English journals).

keywords can be improved

some species are not written in italics (lines 113, 126, 132, 162, 176, 177, 221, 238, 261, 266, 275, 289, 294, 312, 369, 394, 410, 425, 430, 439, 441, 461)

Figure 4: correct "caseintae"

line 279: correct and italicize "Acaccia"

line 302: correct "Ca+2"

Table 3: correct "chease" (twice) and "mozarella"

Which are the best encapsulation methods and materials?

Author Response

  • Author response to report 3:

Author’s notes

Thank you for your comments to improve the manuscript.

keywords can be improved

Thank you for your comment. We changed keyword: „materials” to „viability” this will relate to one of the main aspects of the review which is the viability of bacteria after the microencapsulation proces.

some species are not written in italics (lines 113, 126, 132, 162, 176, 177, 221, 238, 261, 266, 275, 289, 294, 312, 369, 394, 410, 425, 430, 439, 441, 461)

Thank you for your comment. All the names of the microorganisms have been corrected in the manuscript.

Figure 4: correct "caseintae"

Thank you for your comment. We corrected this mistake.

line 279: correct and italicize "Acaccia"

Thank you for your comment. We corrected and italicize this word.

line 302: correct "Ca+2"

Thank you for your comment. We corrected this notation.

Table 3: correct "chease" (twice) and "mozarella"

Thank you for your comment. We corrected this words.

Which are the best encapsulation methods and materials?

After reading the literature on the very broad topic of microencapsulation, it seems that the best methods are those that do not use high temperaturÄ™. It is mainly an extrusion and emulsification method. The most effective materials for probiotics microencapsulating it is definitely a combination of alginate and whey proteins. Studies have shown that they thoroughly secure the core of the microcapsule.

Thank you for all your comments and suggestions. We hope, that the revised version of the manuscript is satisfactory.

Round 2

Reviewer 1 Report

Requested corrections were completed.